# GREPO: A Benchmark for Graph Neural Networks on Repository-Level Bug Localization

## Abstract

Repository-level bug localization, the task of identifying where code must be modified to fix a bug, is a critical software engineering challenge. Standard Large Language Models (LLMs) are often unsuitable for this task due to context window limitations that prevent them from processing entire code repositories. Moreover, the intricate dependencies between code entities mean that bug localization often requires multi-hop reasoning across the repository's structure. Existing approaches typically treat this as an information retrieval (IR) problem, relying on heuristics like keyword matching and text similarity. While some methods incorporate repository graph structures, they often employ simplistic traversal algorithms (e.g., Breadth-First Search). Graph Neural Networks (GNNs) present a promising alternative with their inherent capacity to model complex, repository-wide dependencies, but the absence of a dedicated benchmark has hindered their application. To bridge this gap, we introduce GREPO, the first benchmark designed for repository-scale bug localization using GNNs. It comprises 109 Python repositories and over 10,000 bug-fixing pull requests, offering graph-based data structures ready for direct GNN processing. Our evaluation of various GNN architectures on a representative subset of 9 repositories in GREPO reveals their competitive performance against established information retrieval baselines. This work demonstrates the strong potential of GNNs for this task and establishes GREPO as a foundational resource for future research. Our code can be found at `https://anonymous.4open.science/status/RepoGNN-57C0`.

## 1 Introduction

The ability to accurately and efficiently locate bugs within large code repositories is a fundamental challenge in software engineering. This task, known as repository-level bug localization, is a prerequisite for automatic program repair, enabling developers and automated systems to pinpoint the source of a defect and apply the necessary fix. Böhme et al. (2017) find that professional human developers spend up to 66% of their debugging time on localization, and poor code localization leads to incomplete fixes, new bugs, and significantly extends development cycles.

As modern software repositories can contain millions of lines of code distributed across thousands of files, it is infeasible for humans or Large Language Models (LLMs) to inspect the entire codebase. Moreover, a bug's root cause is often not confined to a single file or function but is instead hidden in a complex, non-local interaction between multiple code entities, requiring multi-hop reasoning across the repository's intricate structure.

Previous works consider bug localization as an information retrieval (IR) task that aims to identify relevant code snippets given natural language descriptions (Xia et al., 2024). Unlike traditional retrieval tasks that primarily focus on lexical or semantic matching between queries and documents, code localization requires multi-hop reasoning. It also necessitates reasoning capabilities to analyze the issue, while considering the structural and semantic properties of code (Chen et al., 2025).

Current approaches to bug localization often fall into two primary categories. The first, and most common, ignores correlation within the code repository and compute similarity to query for each code entities individually. These methods, which include techniques like dense vector similarity, attempt to match a natural language bug description to a relevant piece of code. While effective to

some degree, they are fundamentally limited by their reliance on text similarity and a lack of understanding of the underlying program structure (Lam et al., 2017; Liang et al., 2022). The second category of approaches attempts to model the repository's structure, but these methods are often constrained by the models they employ. For instance, Large Language Models (LLMs) are powerful for code understanding but are hindered by their limited context windows, preventing them from processing an entire repository at once. Furthermore, issue descriptions often mention only symptoms rather than underlying causes, a disconnect that traditional IR struggles to trace across a codebase. Recent LLM-powered agents (Yang et al., 2024; Wang et al., 2024) and other graph-based methods (Liu et al., 2024; Yu et al., 2025) attempt to address this through iterative exploration and simple graph traversal algorithms like Breadth-First Search (BFS) or Monte Carlo Tree Search (MCTS) to perform multi-hop reasoning. These tools, while useful, lack the expressivity and end-to-end learnable capacity needed to fully leverage the repository's complex dependency graph.

Graph Neural Networks (GNNs) offer a promising alternative. With their inherent capacity to learn rich representations of nodes and edges in a graph, GNNs are uniquely suited to model the dependencies between code entities and perform the multi-hop reasoning required for repository-level bug localization. Despite their potential, the application of GNNs to this task has been hindered by a critical gap: the absence of a dedicated benchmark that provides the necessary graph-based data structures for seamless GNN training and evaluation.

To bridge this gap and establish a foundational resource for future research, we introduce **GREPO** (**G**raph **Repo**sitory), the first benchmark specifically designed for repository-level bug localization using GNNs. GREPO offers a comprehensive dataset of 109 Python repositories (full list is in Appendix B) with over 10,000 bug-fixing pull requests, providing a diverse and realistic testbed. Critically, the benchmark provides graph-based data structures that can be processed directly by GNNs. Our main contributions are summarized as follows:

- We introduce GREPO, a new benchmark for repository-level bug localization that provides the first dataset with explicitly defined graph structures ready for GNN processing.
- We detail the collection and pre-processing pipeline for GREPO, which includes 109 open-source Python repositories and over 10,000 bug-fixing pull requests, offering a rich and realistic dataset.
- We evaluate a variety of representative GNN architectures on a representative subset of GREPO and demonstrate their competitive performance against established information retrieval baselines.

## 2 RELATED WORK

### 2.1 BUG LOCALIZATION BENCHMARKS

A bug localization benchmark typically consists of a code repository, bug descriptions, and ground truth labels identifying the location of the fix at various granularities, such as file, class, or function levels. Existing methods on these benchmarks primarily operate by computing a matching score between the bug report and candidate code entities (Niu et al., 2024). Ye et al. (2014) propose most widely used dataset encompassing six open-source Java projects (AspectJ, Birt, Eclipse Platform UI, JDT, SWT, and Tomcat). Other datasets also focus on Java projects (Zhu et al., 2020; Qi et al., 2021; Zou et al., 2021). Datasets on other languages are also available (Sangle et al., 2020; Xiao et al., 2018).

Besides these pure code location datasets, recent bug fixing benchmarks for LLMs and agents also involves bug localization as intermediate step in problem solving. Repo-bench (Liu et al., 2023) and SWE-bench (Jimenez et al., 2024) are two large-scale benchmarks of repo-level codebases code processing, designed to test LLMs for automatic code generation and issue resolution respectively. After that, Zan et al. (2024); Rashid et al. (2025) and Zan et al. (2025) extend to other programming languages. Xu et al. (2025) test models on web development. Yang et al. (2025b) also consider the visual information in software development like syntax highlight or web frameworks.

While various benchmarks for bug localization exist, a dedicated benchmark designed to test Graph Neural Networks (GNNs) on this task is rare. Most datasets lack the explicit graph-based data structures required for direct GNN processing. GREPO fills this gap by providing a plug-and-play benchmark with pre-processed graph inputs, enabling a seamless evaluation of GNNs. Furthermore,

our design allows for easy conversion of location predictions back to text code entities, facilitating direct comparison with other LLM-based methods.

## 2.2 Bug Localization Methods

Early bug localization methods primarily treated the problem as an information retrieval (IR) task, relying on text similarity to identify relevant code. These approaches often used techniques like TF-IDF (Lam et al., 2017) or more advanced text embeddings (Lam et al., 2015; 2017) to match bug reports with source code. More recently, models like CodeBERT (Feng et al., 2020) have been leveraged to learn dense vector representations of both bug reports and code for similarity-based ranking, as seen in FLIM (Liang et al., 2022).

Recognizing the importance of code dependencies, recent research has explored leveraging program graphs, such as Control Flow Graphs (CFGs) and Abstract Syntax Trees (ASTs). Models like GraphCodeBERT (Guo et al., 2021) and UniXCoder (Guo et al., 2022) integrate graph structures with code text to learn more expressive representations. Other approaches, such as Cflow (Zhang et al., 2020), exploit code knowledge graphs. While some works have used GNNs to encode CFGs (Huo et al., 2020; Ma & Li, 2022), these methods have been limited by their failure to incorporate strong language models, resulting in modest performance.

More recently, the rise of AI code agents (Yang et al., 2024; Wang et al., 2024) has shown that bug localization is a critical intermediate step in autonomous bug fixing. These agents typically rely on simple heuristics, such as keyword matching and text similarity, combined with file system commands to navigate repositories. While some advanced agents (Xia et al., 2024) use hierarchical search processes, others have begun incorporating code graphs with simple traversal tools, like Breadth-First Search (BFS) (Chen et al., 2025; Liu et al., 2024; Ouyang et al., 2025) or Monte Carlo Tree Search (MCTS) (Yu et al., 2025). These tools enable multi-hop reasoning, but as the abstract notes, they are often simplistic, not learnable, and lack the expressivity needed to capture complex dependencies across the repository. Our work addresses this limitation by proposing GNNs as a more powerful, learnable alternative to these traversal tools, and our GREPO benchmark provides the necessary foundation for their development and evaluation.

## 3 Preliminaries

**Repository-Level Bug Localization** A bug localization task consists of a code repository of directories and files, a text bug description, and ground truth label identifying the location of the fix at various granularities. In this work, we only consider file and class&function level localization.

**Message Passing Neural Network(MPNN) (Gilmer et al., 2017)** MPNN is a popular GNN framework. It consists of multiple message-passing layers, where the $k$-th layer is:

$$\boldsymbol{h}_v^{(k)} = U^{(k)}(\boldsymbol{h}_v^{(k-1)}, \text{AGG}(\{M^{(k)}(\boldsymbol{h}_u^{(k-1)}) \mid u \in V, (u,v) \in E\})), \tag{1}$$

where $\boldsymbol{h}_v^{(k)}$ is the representation of node $v$ at the $k$-th layer, $U^{(k)}$ and $M^{(k)}$ are functions such as Multi-Layer Perceptrons (MLPs), and AGG is an aggregation function like sum or max. The initial node representation $\boldsymbol{h}_v^{(0)}$ is the node feature $X_v$. Each layer aggregates information from neighbors to update the center node's representation.

## 4 Dataset Construction

Our objective is to construct a dataset of repository graphs suitable for evaluating both vanilla Graph Neural Networks (which utilize feature vectors as node inputs) and Graph Large Language Models (which can process raw text). The dataset is designed to facilitate the task of repository-level bug location. Figure 1 provides an overview of our construction pipeline, which consists of three primary stages: (1) converting repositories into a temporal graph structure where each commit represents a snapshot (Section 4.1), (2) collecting and filtering pull requests and issues to establish ground-truth labels for the bug location task (Section 4.2), and (3) generating feature embeddings from code text to equip the graph nodes with semantic information, including the creation of an augmented query embedding for the bug description (Section 4.3).

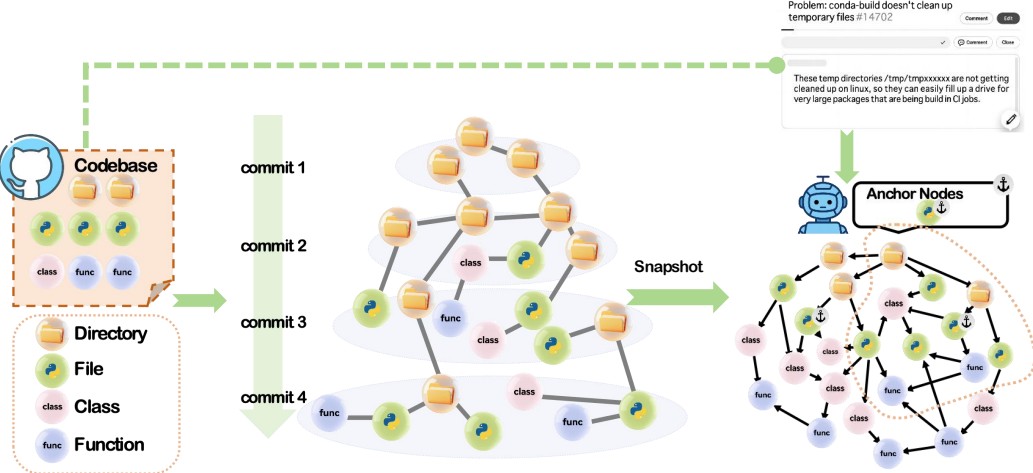

Figure 1: An overview of the dataset construction pipeline, which consists of three core stages: (1) converting code repositories into a temporal graph structure with incremental updates, (2) collecting and filtering pull requests and issues to derive bug location labels, and (3) generating semantic feature embeddings for graph nodes and bug descriptions, and then utilizing these embeddings along with LLM to extract suspicious nodes, which are referred to as anchor nodes. For each anchor node, we extract a K-hop subgraph centered at it and run GNN on the subgraph.

## 4.1 BUILDING CODE REPOSITORY GRAPH

A code repository is a dynamic entity that evolves over time. As each bug location task is associated with a specific historical state of the repository (i.e., a particular commit), it is necessary to model the repository's graph structure at different points in time. Building an individual graph for each commit is computationally infeasible due to the sheer number of commits. To address this, we construct a single temporal graph where each node is annotated with start and end timestamps denoting its period of validity. This design allows nodes corresponding to unchanged files to be shared across multiple commits, enabling efficient, incremental graph construction.

**Graph of One Commit.** For any given commit, we parse the repository structure into a heterogeneous graph comprising four node types: Directory, File, Function, and Class. We establish the following directed edge types to capture the complex relationships within the codebase:

- Contain (and its reverse, ContainedIn): Represents the hierarchical structure (e.g., Directory to File, File to Class, Class to Function). These relations are parsed using the Python AST library, forming a tree structure.

- Call (and Called): Represents a function or method invoking another.

- Child (and Parent): Represents class inheritance. The Call and Child relations are parsed using the Jedi library, which performs static analysis to infer these dependencies.

**Temporal Relation Between Commits.** We utilize the GitHub API to collect the complete history of a repository, including all commits, pull requests (PRs), and issues. The commit history naturally forms a Directed Acyclic Graph (DAG) due to branching and merging. However, as most temporal models are designed for linear sequences, we linearize this history by extracting the longest path in the commit DAG that contains the commit of interest for a given task. For training data, we use the global longest path in the repository's history. For testing, we use the longest path that terminates at the commit associated with the reported bug. This approach is justified as the development history in well-maintained repositories is predominantly linear; in our dataset, over 75% of commits in the selected repositories reside in the longest path of their main/master branch, which we focus on as it contains the stable, canonical code evolution.

**Incremental Building.** Our temporal graph is built incrementally to maximize efficiency. Each node is created with a start timestamp. For a new commit, we parse the associated patch file to identify changed files. Only the AST and relational edges (Call, Child) for these changed files are reparsed. New nodes are created for added entities, and the end timestamp of removed entities is set. Reverse edges (e.g., CalledBy) are added in a post-processing step. This incremental approach avoids the cost of rebuilding the entire graph for each commit.

## 4.2 TASK AND LABEL COLLECTION

The core task for our benchmark is, given a bug report (issue description), to predict the set of graph nodes (files, classes, and/or functions) that require modification to fix the bug.

To construct high-quality labels for this task, we collect closed and merged pull requests (PRs) that modify Python files, ensuring the fixes are valid and incorporated into the codebase. We then link these PRs to their corresponding GitHub issues using regular expressions to match issue numbers in PR titles and descriptions. The first message (initial description) of the linked issue serves as the bug description text; we deliberately exclude subsequent discussion and the PR's fix description to prevent data leakage that would trivialize the learning task.

To enhance the precision of our labels and the bug descriptions, we employ an LLM-based extraction step. The LLM is prompted to parse the raw issue text into structured components, such as a general description, environment setup, reproduction script, error traceback, and reason guess by the reporter. This structured data provides a cleaner signal for model training and evaluation.

The ground-truth labels for the location task are the set of files, classes, and functions that were actually modified in the linked, fixing PR. This provides a precise, objective measure of success for models.

## 4.3 GRAPH FEATURE CONSTRUCTION

To equip the graph for model processing, we generate feature representations for all text-bearing nodes (e.g., File, Class, Function nodes contain code; Issue nodes contain text).

**Text Embedding for GNNs** For vanilla GNNs that require fixed-size feature vectors, we encode the textual content of each node (e.g., function code, class code, file path) into a dense vector representation using the Qwen-Embedding model. This model provides high-quality, semantic embeddings suitable for capturing code semantics.

**Anchor Node and Query Augmentation** Following methodologies inspired by CodeCGM, we augment the graph with a special anchor node for each bug location task. This node is connected to the nodes containing the text of the issue report. The text of the bug description is rewritten by an LLM to be more concise and structured, and is then encoded into a bug description embedding using the same Qwen-Embedding model.

**Similarity Feature** To provide a strong initial signal for retrieval, we compute the inner product (cosine similarity) between the bug description embedding and the text embedding of every other node in the graph. This similarity score is added as an initial node feature, directly guiding the model towards textually relevant entities and easing the learning process. Moreover, similarity is low-dimension and valid cross different repositories.

## 5 THE GREPO GRAPH FORMULATION

**Snapshot Selection** We treat discrete commit IDs as discrete time points, denoted as $t$. For a given pull request $pr_i$, which is associated with one or more related or resolved issues $\{iss\}_i$, the PR has a base commit. This commit inherently contains the bug that $pr_i$ aims to fix, and we denote this specific time point as $t_{bug}$. Since $t_{bug}$ is a necessary condition for the bug's existence, any graph structure from a time $t > t_{bug}$ must be excluded during the localization process for the bug described in $\{iss\}_i$ to prevent information leakage from subsequent code changes that might contain hints about the fix.

Each node in the graph is annotated with start and end timestamps indicating its period of validity. To construct the snapshot of the repository graph at $t_{bug}$, we filter the nodes as follows:

$$V_{t_{bug}} = \{v \in V | t_{start}(v) \leq t_{bug} \leq t_{end}(v)\}$$

The corresponding edge set for this snapshot is naturally derived from the filtered nodes:

$$E_i(K) = \{(u, v, r) \in E_{t_{bug}} | u, v \in V_i(K)\},$$

where $u$ and $v$ are nodes, and $r$ denotes the type of the edge $(u, v)$.

**Graph Characteristics**   Edge types are critical for understanding the code structure. We provide these types as edge attributes ( edge_attr ) to the GNN. The specific types include: contain, reverse contain, previous, next, superclasses, child class, call, and called. A detailed analysis of the impact of edge types on GNN performance will be presented in 6.5.

Node features are constructed by leveraging three key pieces of information inherent to each node: its path (reflecting its topological position within the graph), its name (serving as the most direct retrievable identifier), and its content (encapsulating the detailed code or documentation). Furthermore, we incorporate the semantic information from the issue description itself (comments are carefully excluded to prevent data leakage). The textual information from the node (path, name, content) and the issue description are encoded into embeddings using the Qwen-Embedding model. The inner product between these embeddings is used as a primary node feature. This approach is motivated by two factors: firstly, the inner product inherently represents semantic similarity, enabling the direct identification of highly suspicious candidate nodes; secondly, the high dimensionality of the raw Qwen-Embedding output poses challenges for GNN training, and using the inner product effectively reduces the feature dimensionality.

Inspired by CodeCGM (Tao et al., 2025a), we also leverage the advanced natural language capabilities of LLMs to directly match relevant keywords and phrases from the issue description against node names and paths, generating an additional set of candidate nodes. This step is crucial because converting text to embeddings can lead to information loss, and issue reports often contain specific error traces or code snippets that can be directly matched to specific file or function names, providing a strong starting point for localization. The union of nodes selected via embedding similarity and those retrieved via direct name/path matching constitutes the set of anchor nodes. A binary feature indicating whether a node is an anchor node is concatenated with the similarity feature, serving as a direct indicator of the node's initial suspiciousness.

**Extracting Subgraphs Around Anchor Nodes**   Bugs often exhibit locality; we hypothesize that anchor nodes provide a strong initial indication of the bug's vicinity, and the actual faulty code entities and modification sites are likely located within the $k$-hop neighborhood of these anchors. Furthermore, for computational efficiency, running GNNs on $k$-hop subgraphs centered at anchor nodes is a practical necessity. Therefore, for each anchor node, we extract a $k$-hop subgraph from the snapshot graph $(V_{t_{bug}}, E_{t_{bug}})$:

$$V_i(K) = \{v \in V_{t_{bug}} | d(a_i, v) \leq K\}, E_i(K) = \{(u, v, r) \in E_{t_{bug}} | u, v \in V_i(K)\},$$

where $a_i$ is an anchor node, $u$ and $v$ are nodes, $r$ is the edge type, and $d(a_i, v)$ denotes the shortest path distance between $a_i$ and $v$ in the graph.

In our experiments, we observed that using 1-hop or 2-hop subgraphs ($K = 1$ or $K = 2$) around anchor nodes was sufficient to encompass the majority (on average, over 80%) of the nodes that ultimately required modification.

Table 1: Main Results: GNNs vs. Baselines on Bug Localization.

| Method | File-Level | | | | Class&Function-Level | | | |
|---|---|---|---|---|---|---|---|---|
| Metric | Hit@1 | Hit@5 | Hit@10 | Hit@20 | Hit@1 | Hit@5 | Hit@10 | Hit@20 |
| LocAgent | $26.85_{\pm16.06}$ | $35.25_{\pm19.62}$ | $35.86_{\pm19.26}$ | $35.86_{\pm19.26}$ | $3.19_{\pm3.28}$ | $14.21_{\pm12.21}$ | $16.80_{\pm13.29}$ | $17.03_{\pm13.25}$ |
| AgentLess | $92.72_{\pm4.81}$ | $93.00_{\pm4.32}$ | $93.00_{\pm4.32}$ | $93.00_{\pm4.32}$ | $6.26_{\pm3.64}$ | $21.58_{\pm6.33}$ | $21.81_{\pm6.70}$ | $21.81_{\pm6.70}$ |
| CFRAG | $2.09_{\pm1.15}$ | $10.92_{\pm5.34}$ | $23.07_{\pm10.79}$ | $34.74_{\pm14.13}$ | $0.47_{\pm0.30}$ | $3.77_{\pm2.43}$ | $7.79_{\pm4.84}$ | $12.12_{\pm7.58}$ |
| UniMP | $19.32_{\pm7.32}$ | $62.99_{\pm10.43}$ | $80.76_{\pm7.65}$ | $93.11_{\pm3.29}$ | $2.64_{\pm2.99}$ | $10.74_{\pm7.71}$ | $20.28_{\pm13.49}$ | $38.75_{\pm15.83}$ |
| GPS | $36.47_{\pm7.03}$ | $80.73_{\pm6.06}$ | $92.10_{\pm4.09}$ | $96.88_{\pm2.31}$ | $10.86_{\pm4.14}$ | $34.16_{\pm11.40}$ | $51.83_{\pm12.93}$ | $63.95_{\pm11.96}$ |
| GIN | $49.02_{\pm11.60}$ | $87.88_{\pm9.54}$ | $94.65_{\pm6.54}$ | $98.84_{\pm1.15}$ | $3.74_{\pm2.81}$ | $15.93_{\pm8.56}$ | $28.89_{\pm11.39}$ | $42.57_{\pm14.25}$ |
| SAGE | $51.87_{\pm11.55}$ | $88.03_{\pm7.43}$ | $95.53_{\pm3.21}$ | $98.70_{\pm1.14}$ | $4.61_{\pm2.30}$ | $16.15_{\pm5.03}$ | $28.11_{\pm8.28}$ | $46.42_{\pm13.35}$ |
| GAT | $54.18_{\pm9.07}$ | $87.95_{\pm7.27}$ | $95.04_{\pm3.95}$ | $98.59_{\pm0.88}$ | $22.27_{\pm9.54}$ | $52.89_{\pm12.17}$ | $66.39_{\pm11.05}$ | $75.82_{\pm8.13}$ |

Table 2: Cross-Repository Transfer Learning Analysis.

| File | astropy | | dvc | | ipython | | pylint | |
|---|---|---|---|---|---|---|---|---|
| | File | Func | File | Func | File | Func | File | Func |
| Joint | 86.91 | 43.79 | 74.12 | 55.13 | 95.71 | 50.41 | 83.33 | 54.46 |
| Individual | 48.33 | 33.55 | 31.93 | 37.00 | 55.25 | 27.76 | 34.83 | 27.66 |
| Transfer | 91.95 | 42.58 | 72.73 | 57.73 | 94.42 | 57.12 | 89.23 | 55.65 |
| | 84.69 | 41.41 | 74.32 | 39.40 | 94.81 | 49.74 | 83.37 | 57.98 |
| | 84.46 | 41.46 | 74.57 | 40.95 | 82.50 | 43.85 | 88.22 | 57.67 |
| | 86.81 | 41.62 | 84.71 | 40.88 | 85.63 | 46.24 | 78.03 | 45.19 |

# 6 EXPERIMENTS

## 6.1 EXPERIMENTAL SETUP

To comprehensively evaluate the performance and generalization capability of our proposed approach, we conduct a series of experiments on a curated subset of the GREPO dataset. Specifically, we select nine representative open-source software repositories spanning diverse domains and scales. These repositories—astropy, dvc, ipython, pylint, scipy, sphinx, streamlink, xarray, and geopandas—were chosen to ensure a varied evaluation benchmark in terms of application domain, project size, and contribution activity. For each repository, we partition the pull requests chronologically to simulate a realistic deployment scenario: the earliest 80% of pull requests form the training set, the most recent 20% are held out as the test set. This temporal split mitigates data leakage and provides a realistic assessment of the model's ability to generalize to future project states.

Text embeddings for both code snippets and natural language queries are generated using the Qwen3-Embedding-8B model (Zhang et al., 2025), which provides high-quality semantic representations crucial for capturing the nuances of developer intent and code semantics.

Our experimental evaluation is designed to address three key research questions:

1. **Effectiveness of GNN Methods:** We compare our GNN-based approach against traditional information retrieval (IR) baselines to verify the superiority of leveraging graph structural information for code localization.

2. **Cross-Repository Transferability:** We investigate the model's capacity to generalize across different projects by comparing three training paradigms: (a) models jointly trained on all 9 repositories, (b) models individually trained on each single repository, and (c) models trained on a subset of 6 repositories and directly tested on the remaining 3. This setup assesses the model's ability to transfer knowledge without repository-specific fine-tuning.

3. **Feature Ablation Studies:** We perform an ablation analysis to dissect the contribution of each feature component in our model, providing insights into the relative importance of different information sources.

## 6.2 EVALUATION METRICS

Following established practices in code retrieval and localization (Chen et al., 2025), we adopt **Hit@k** (Hit ratio at k samples) as our primary evaluation metric. For a given query, the model returns a ranked list of top-k candidate code locations. The Hit@k score is computed as the size of the intersection between the set of predicted top-k locations and the ground-truth set of required modifications, normalized by the minimum of k and the number of ground-truth locations. This metric effectively measures the model's ability to identify all necessary code sections that require changes within a limited budget of k suggestions. We report results for multiple values of k: 1, 5, 10, and 20, providing a comprehensive view of the model's performance across different practical constraints. Furthermore, evaluations are conducted at two granularity levels: **file-level localization** and **class/function-level localization**, addressing different stages of the code change process.

## 6.3 COMPARISON WITH INFORMATION RETRIEVAL METHODS

To comprehensively evaluate the performance of our proposed GNN approach, we compare against several representative information retrieval baselines. These include: (1) **LocAgent** (Chen et al., 2025), which utilizes the powerful Qwen2.5-72B-Instruct model (Yang et al., 2025a) in an agent-based framework; (2) **Agentless** (Xia et al., 2024), which employs GPT-4o (Hurst et al., 2024) for direct code retrieval without agent components; and (3) **CF-RAG**, the retrieval-augmented generation approach from CodeFuse (Tao et al., 2025b) that uses Qwen3-Embedding-8B (Zhang et al., 2025) for semantic similarity matching. Notably, CF-RAG also serves as the method for generating the anchor nodes used in our GNN approach, allowing us to isolate the contribution of the graph neural network itself rather than merely the quality of the initial retrieval.

All baseline methods are evaluated without additional training, applying them directly to the test sets of each repository. For the GNN comparisons, we benchmark several well-established graph neural network architectures including UniMP (Shi et al., 2021), GIN (Xu et al., 2019), GraphSage (Hamilton et al., 2017), the graph transformer GPS (Rampásek et al., 2022), and GAT (Velickovic et al., 2017). All GNN models are jointly trained on the training sets from all nine repositories to ensure consistent comparison conditions.

The experimental results, presented in Table 1, report the mean test scores across all nine repositories along with their standard deviations. The raw results on each repository are in Appendix C. Our findings demonstrate that GAT achieves the best performance among all evaluated GNN architectures. Consequently, we adopt GAT for all subsequent experiments. More importantly, the results show that GPS, GIN, GraphSage, and GAT consistently outperform both LocAgent and CF-RAG across all datasets, validating the effectiveness of incorporating graph structural information through GNNs for code localization tasks. Remarkably, GAT even significantly surpasses the Agentless approach with GPT-4o at the Class&Function-Level localization task, suggesting that the inductive biases introduced by graph neural networks capture code structural patterns that even powerful large language models may miss without explicit graph reasoning.

## 6.4 CROSS-REPOSITORY TRANSFERABILITY ANALYSIS

In this section, we employ the GAT architecture with identical hyperparameters to investigate the transferability of our approach across different software repositories. Our primary configuration involves training the GNN jointly on all nine repositories. To systematically evaluate cross-repository transfer capability, we introduce two additional experimental settings:

1. **Individual Training:** A separate GNN model is trained exclusively on each repository's training set and evaluated on the corresponding test set. This setting establishes a baseline for repository-specific performance without knowledge transfer.

2. **Transfer Training:** Models are trained jointly on a subset of six repositories and directly tested on the three held-out repositories without any fine-tuning. This setting rigorously assesses the model's ability to generalize to completely unseen projects.

The results of these experiments are summarized in Table 2, where blue cells indicate repositories that were excluded from training and used only for testing. Two key observations emerge from

Table 3: Component-wise Ablation Analysis.

| Method | File-Level | | | | Class&Function-Level | | | |
|---|---|---|---|---|---|---|---|---|
| Metric | Hit@1 | Hit@5 | Hit@10 | Hit@20 | Hit@1 | Hit@5 | Hit@10 | Hit@20 |
| Full | $54.18_{\pm9.07}$ | $87.95_{\pm7.27}$ | $95.04_{\pm3.95}$ | $98.59_{\pm0.88}$ | $22.27_{\pm9.54}$ | $52.89_{\pm12.17}$ | $66.39_{\pm11.05}$ | $75.82_{\pm8.13}$ |
| w/o contain | $34.37_{\pm10.54}$ | $58.15_{\pm10.81}$ | $69.72_{\pm10.42}$ | $77.65_{\pm7.09}$ | $6.76_{\pm5.61}$ | $22.03_{\pm10.08}$ | $35.06_{\pm10.80}$ | $49.87_{\pm11.52}$ |
| w/o call | $31.36_{\pm10.64}$ | $68.00_{\pm10.90}$ | $82.18_{\pm8.30}$ | $88.83_{\pm5.73}$ | $9.02_{\pm6.22}$ | $28.75_{\pm10.65}$ | $43.30_{\pm10.63}$ | $56.18_{\pm9.50}$ |
| w/o inherit | $23.47_{\pm7.77}$ | $61.65_{\pm12.23}$ | $77.02_{\pm6.33}$ | $85.18_{\pm4.76}$ | $7.54_{\pm7.64}$ | $24.93_{\pm11.46}$ | $39.34_{\pm12.60}$ | $51.23_{\pm12.84}$ |
| w/o Edge | $3.82_{\pm3.17}$ | $21.14_{\pm7.45}$ | $43.94_{\pm12.44}$ | $68.17_{\pm10.27}$ | $6.74_{\pm5.85}$ | $20.98_{\pm10.37}$ | $35.07_{\pm12.02}$ | $47.52_{\pm13.39}$ |
| w/o sim | $4.11_{\pm1.90}$ | $18.91_{\pm11.91}$ | $40.68_{\pm10.91}$ | $71.43_{\pm7.24}$ | $0.44_{\pm0.44}$ | $1.90_{\pm0.97}$ | $3.91_{\pm1.71}$ | $8.76_{\pm2.27}$ |
| w/o anchor | $9.48_{\pm3.49}$ | $36.60_{\pm7.35}$ | $53.33_{\pm9.53}$ | $78.00_{\pm8.06}$ | $6.23_{\pm4.25}$ | $21.04_{\pm8.60}$ | $34.01_{\pm10.59}$ | $48.35_{\pm11.54}$ |
| w/o Node | $0.80_{\pm1.01}$ | $4.10_{\pm2.13}$ | $8.05_{\pm3.11}$ | $11.75_{\pm3.87}$ | $0.88_{\pm0.90}$ | $2.70_{\pm0.90}$ | $3.62_{\pm1.25}$ | $4.64_{\pm1.65}$ |

the analysis: First, joint training consistently outperforms individual training across all repositories, demonstrating that learning from multiple projects simultaneously enhances performance on each individual repository through beneficial knowledge transfer. Second, in the direct transfer setting, the model achieves comparable or even superior performance on the held-out repositories relative to models trained jointly on all datasets, highlighting the strong generalization capability and cross-repository transferability of our GNN approach. These findings suggest that the graph neural network effectively learns universal patterns of code change localization that transcend project-specific characteristics.

### 6.5 ABLATION

In section, we use GAT and all the same hyperparameters. We ablate the input features of GNN to verify the effectiveness of these features. In Table 3, full denote full model. To analyse important of node feature, we conduct experiments: w/o contain, w/o call, and w/o inherit mean we remove contain, call, and inherit type edge and its reverse type edge from input graph, w/o Edge denote keep all edge but do not use edge type. To analyse important of node feature, we conduct experiments: w/o sim, w/o anchor means we remove node text-query similar, anchor node label from input graph. w/o node means do not use node feature. All ablation leads to significant performance drop, verifying the effectiveness of all our node and edge features.

## 7 CONCLUSION

Repository-level bug localization remains a critical and challenging task in software engineering, demanding both an understanding of natural language bug descriptions and the ability to reason over the complex, interconnected structure of large codebases. This paper introduces GREPO, the first benchmark designed for Graph Neural Networks (GNNs) to do tasks on repository-level bug localization. By providing a dataset of 109 Python repositories with pre-processed graph structures, GREPO enables the direct application of GNNs to this critical software engineering task. Our experiments demonstrate that GNNs, effectively leverage code structure to outperform traditional information retrieval baselines, showcasing strong performance and cross-repository generalizability. GREPO establishes a new foundation for research into structural reasoning for code understanding, and we release it to the community to foster further innovation.

## 8 LIMITATION

We have actually utilized the results of GNN to test the Agent on SWE-Bench-Live, but the outcomes were unsatisfactory. Empirical studies indicate that while GNN performs exceptionally well on the tasks in SWE-Bench-Live, its accuracy in localization does not significantly contribute to improving the Agent's localization accuracy or issue resolution rate. We believe this is due to the Agent's weak ability to follow instructions. In future research, we will focus on how to organically and efficiently integrate GNN and the Agent.

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

## A  THE USE OF LARGE LANGUAGE MODELS (LLMS)

We use an LLM to polish the sentence structures and word choices in our paper, as well as to check for grammar errors.

## B  THE REPOSITORIES IN GREPO

The full repository names of the GREPO benchmark are shown in Table 4.

## C  DETAILED RESULTS OF VARIOUS METHODS

The detailed results of multiple methods, including our two GNN training approaches, are presented in the following tables: 5, 6, 7, 8, 9, and 10. The meaning of Single-Train is the results obtained by training on a single repository respectively, while Joint-Train refers to the results obtained by joint training on these 9 repositories.

Table 4: The full repository names of the GREPO benchmark

| Cirq | Fast-F1 | Flexget | LLaMA-Factory |
|---|---|---|---|
| PyBaMM | PyPSA | Radicale | Solaar |
| WeasyPrint | aider | aiogram | ansible-lint |
| arviz | astroid | astropy | attrs |
| azure-sdk-for-python | babel | beancount | beets |
| briefcase | buku | cfn-lint | checkov |
| conan | conda | crawlee-python | cryptography |
| csvkit | datamodel-code-generator | datasets | django |
| dspy | dvc | dynaconf | faker |
| falcon | fastmcp | faststream | feature_engine |
| filesystem_spec | flask | fonttools | fusesoc |
| geopandas | gitingest | haystack | icloud_photos_downloader |
| instructlab | ipython | jax | jupyter-ai |
| kedro | keras | kirara-ai | linkding |
| litellm | llama-stack | llama_deploy | loguru |
| marshmallow | matplotlib | mcp-atlassian | mesa |
| mypy | networkx | ntc-templates | openai-agents-python |
| patroni | pdm | pipenv | poetry |
| privacyidea | pvlib-python | pydicom | pylint |
| pymdown-extensions | pyomo | python | python-control |
| python-telegram-bot | pyvista | qtile | reflex |
| scipy | scrapy-splash | segmentation_models.pytorch | sh |
| shapely | smart_open | smolagents | sphinx |
| sqlfluff | sqllineage | starlette | streamlink |
| supervisor | tablib | textual | torchtune |
| tox | transformers | transitions | trimesh |
| twine | urllib3 | wemake-python-styleguide | xarray |
| yt-dlp | | | |

Table 5: Performance results of various types of methods at the file granularity level.

| Single-Train | astropy | dvc | ipython | pylint | scipy | sphinx | streamlink | xarray | geopandas |
|---|---|---|---|---|---|---|---|---|---|
| **Hit@1** | 0.1915 | 0.1292 | 0.2647 | 0.1687 | 0.1469 | 0.2989 | 0.3152 | 0.1309 | 0.2402 |
| **Hit@5** | 0.4833 | 0.3193 | 0.5525 | 0.3483 | 0.5028 | 0.5436 | 0.7495 | 0.3593 | 0.4578 |
| **Hit@10** | 0.5448 | 0.3925 | 0.6392 | 0.4118 | 0.5917 | 0.6192 | 0.8076 | 0.4354 | 0.5261 |
| **Hit@20** | 0.5865 | 0.4425 | 0.6789 | 0.5039 | 0.6423 | 0.6778 | 0.8556 | 0.5451 | 0.5456 |
| **Joint-Train** | astropy | dvc | ipython | pylint | scipy | sphinx | streamlink | xarray | geopandas |
| **Hit@1** | 0.4873 | 0.4457 | 0.5439 | 0.5673 | 0.5728 | 0.6389 | 0.7079 | 0.4409 | 0.4722 |
| **Hit@5** | 0.8691 | 0.7412 | 0.9571 | 0.8333 | 0.9242 | 0.9167 | 0.9749 | 0.8315 | 0.8681 |
| **Hit@10** | 0.9458 | 0.8687 | 0.9769 | 0.9074 | 0.9732 | 0.9750 | 0.9910 | 0.9437 | 0.9722 |
| **Hit@20** | 0.9763 | 0.9798 | 0.9872 | 0.9747 | 0.9902 | 1.0000 | 0.9946 | 0.9912 | 0.9792 |
| **CF-RAG** | astropy | dvc | ipython | pylint | scipy | sphinx | streamlink | xarray | geopandas |
| **Hit@1** | 0.0238 | 0.0170 | 0.0185 | 0.0066 | 0.0331 | 0.0066 | 0.0187 | 0.0214 | 0.0423 |
| **Hit@5** | 0.0856 | 0.0491 | 0.1451 | 0.0538 | 0.1579 | 0.0645 | 0.2017 | 0.0868 | 0.1380 |
| **Hit@10** | 0.2098 | 0.1336 | 0.3532 | 0.1040 | 0.2903 | 0.1672 | 0.4260 | 0.1509 | 0.2411 |
| **Hit@20** | 0.3252 | 0.2254 | 0.4716 | 0.1652 | 0.4652 | 0.2652 | 0.6039 | 0.2516 | 0.3530 |
| **Locagent** | astropy | dvc | ipython | pylint | scipy | sphinx | streamlink | xarray | geopandas |
| **Hit@1** | 0.2708 | 0.1517 | 0.6042 | 0.0404 | 0.3422 | 0.1647 | 0.3592 | 0.2087 | 0.2750 |
| **Hit@5** | 0.3542 | 0.2450 | 0.8042 | 0.0825 | 0.4174 | 0.2517 | 0.3908 | 0.2936 | 0.3333 |
| **Hit@10** | 0.3625 | 0.2450 | 0.8042 | 0.0925 | 0.4174 | 0.2796 | 0.3908 | 0.3027 | 0.3333 |
| **Hit@20** | 0.3625 | 0.2450 | 0.8042 | 0.0925 | 0.4174 | 0.2796 | 0.3908 | 0.3027 | 0.3333 |
| **Agentless** | astropy | dvc | ipython | pylint | scipy | sphinx | streamlink | xarray | geopandas |
| **Hit@1** | 0.9444 | 0.9091 | 0.9792 | 0.9333 | 0.8462 | 0.8684 | 0.9111 | 0.9762 | 0.9773 |
| **Hit@5** | 0.9444 | 0.9091 | 0.9792 | 0.9333 | 0.8718 | 0.8684 | 0.9111 | 0.9762 | 0.9773 |
| **Hit@10** | 0.9444 | 0.9091 | 0.9792 | 0.9333 | 0.8718 | 0.8684 | 0.9111 | 0.9762 | 0.9773 |
| **Hit@20** | 0.9444 | 0.9091 | 0.9792 | 0.9333 | 0.8718 | 0.8684 | 0.9111 | 0.9762 | 0.9773 |

Table 6: Joint-training Results and Cross-Repository Transfer Learning Analysis at the file granularity level.

| Joint-Type | | astropy | dvc | ipython | pylint | scipy | sphinx | streamlink | xarray | geopandas |
|---|---|---|---|---|---|---|---|---|---|---|
| | **Hit@1** | 0.4873 | 0.4457 | 0.5439 | 0.5673 | 0.5728 | 0.6389 | 0.7079 | 0.4409 | 0.4722 |
| Joint-ALL | **Hit@5** | 0.8691 | 0.7412 | 0.9571 | 0.8333 | 0.9242 | 0.9167 | 0.9749 | 0.8315 | 0.8681 |
| | **Hit@10** | 0.9458 | 0.8687 | 0.9769 | 0.9074 | 0.9732 | 0.9750 | 0.9910 | 0.9437 | 0.9722 |
| | **Hit@20** | 0.9763 | 0.9798 | 0.9872 | 0.9747 | 0.9902 | 1.0000 | 0.9946 | 0.9912 | 0.9792 |
| | **Hit@1** | 0.5532 | 0.4028 | 0.6208 | 0.6178 | 0.5777 | 0.6222 | 0.7670 | 0.4344 | 0.5347 |
| -w/o astropy | **Hit@5** | 0.9195 | 0.7273 | 0.9442 | 0.8923 | 0.9298 | 0.9583 | 0.9857 | 0.8775 | 0.9358 |
| | **Hit@10** | 0.9778 | 0.8384 | 0.9679 | 0.9461 | 0.9667 | 0.9792 | 0.9857 | 0.9537 | 0.9688 |
| | **Hit@20** | 0.9890 | 0.9697 | 1.0000 | 0.9798 | 0.9907 | 0.9833 | 0.9946 | 0.9912 | 0.9896 |
| | **Hit@1** | 0.4921 | 0.4320 | 0.5920 | 0.5976 | 0.5915 | 0.6389 | 0.7563 | 0.4325 | 0.4688 |
| -w/o astropy | **Hit@5** | 0.8469 | 0.7432 | 0.9481 | 0.8337 | 0.9246 | 0.8833 | 0.9749 | 0.8363 | 0.8750 |
| dvc | **Hit@10** | 0.9266 | 0.8491 | 0.9615 | 0.9562 | 0.9695 | 0.9667 | 0.9910 | 0.9358 | 0.9358 |
| | **Hit@20** | 0.9752 | 0.9376 | 0.9872 | 0.9798 | 0.9889 | 1.0000 | 0.9946 | 0.9888 | 0.9583 |
| | **Hit@1** | 0.4929 | 0.4190 | 0.4403 | 0.5118 | 0.5774 | 0.5806 | 0.7509 | 0.4427 | 0.5035 |
| -w/o astropy | **Hit@5** | 0.8446 | 0.7457 | 0.8250 | 0.8822 | 0.9227 | 0.9250 | 0.9713 | 0.9005 | 0.9097 |
| dvc | **Hit@10** | 0.9435 | 0.8337 | 0.9346 | 0.9512 | 0.9708 | 0.9667 | 0.9910 | 0.9574 | 0.9514 |
| ipython | **Hit@20** | 0.9848 | 0.9496 | 0.9922 | 0.9848 | 0.9916 | 1.0000 | 1.0000 | 1.0000 | 0.9583 |
| | **Hit@1** | 0.5068 | 0.4705 | 0.4730 | 0.4342 | 0.5615 | 0.6139 | 0.6810 | 0.4372 | 0.4826 |
| -w/o astropy | **Hit@5** | 0.8681 | 0.8471 | 0.8563 | 0.7803 | 0.8663 | 0.9417 | 0.9749 | 0.8289 | 0.8646 |
| dvc | **Hit@10** | 0.9158 | 0.9304 | 0.9525 | 0.9025 | 0.9665 | 0.9750 | 0.9892 | 0.9256 | 0.9462 |
| ipython pylint | **Hit@20** | 0.9813 | 0.9713 | 0.9888 | 0.9714 | 0.9841 | 1.0000 | 1.0000 | 0.9944 | 0.9722 |

Table 7: Performance results of various types of methods at the Func/Class granularity level.

| Single-Train | astropy | dvc | ipython | pylint | scipy | sphinx | streamlink | xarray | geopandas |
|---|---|---|---|---|---|---|---|---|---|
| **Hit@1** | 0.1160 | 0.1091 | 0.1125 | 0.1347 | 0.1130 | 0.1113 | 0.1050 | 0.1154 | 0.0729 |
| **Hit@5** | 0.3355 | 0.3700 | 0.2776 | 0.2766 | 0.2934 | 0.3049 | 0.3958 | 0.3770 | 0.2976 |
| **Hit@10** | 0.4250 | 0.5040 | 0.4345 | 0.3483 | 0.3517 | 0.5092 | 0.5835 | 0.4625 | 0.3288 |
| **Hit@20** | 0.4986 | 0.6472 | 0.6164 | 0.4084 | 0.4510 | 0.5356 | 0.7028 | 0.5298 | 0.3547 |

| Joint-Train | astropy | dvc | ipython | pylint | scipy | sphinx | streamlink | xarray | geopandas |
|---|---|---|---|---|---|---|---|---|---|
| **Hit@1** | 0.1796 | 0.2442 | 0.2178 | 0.2646 | 0.2647 | 0.1626 | 0.4276 | 0.1435 | 0.0999 |
| **Hit@5** | 0.4379 | 0.5513 | 0.5041 | 0.5446 | 0.5603 | 0.5088 | 0.8170 | 0.4225 | 0.4140 |
| **Hit@10** | 0.5953 | 0.6199 | 0.7175 | 0.6877 | 0.7272 | 0.6239 | 0.9012 | 0.5389 | 0.5635 |
| **Hit@20** | 0.7268 | 0.7774 | 0.7858 | 0.7396 | 0.7949 | 0.6822 | 0.9408 | 0.6809 | 0.6955 |

| CF-RAG | astropy | dvc | ipython | pylint | scipy | sphinx | streamlink | xarray | geopandas |
|---|---|---|---|---|---|---|---|---|---|
| **Hit@1** | 0.0064 | 0.0000 | 0.0081 | 0.0009 | 0.0026 | 0.0050 | 0.0052 | 0.0057 | 0.0087 |
| **Hit@5** | 0.0344 | 0.0309 | 0.0252 | 0.0130 | 0.0712 | 0.0132 | 0.0834 | 0.0284 | 0.0397 |
| **Hit@10** | 0.0742 | 0.0571 | 0.0497 | 0.0409 | 0.1513 | 0.0288 | 0.1675 | 0.0613 | 0.0701 |
| **Hit@20** | 0.1339 | 0.0811 | 0.0627 | 0.0746 | 0.2598 | 0.0551 | 0.2369 | 0.0886 | 0.0980 |

| Locagent | astropy | dvc | ipython | pylint | scipy | sphinx | streamlink | xarray | geopandas |
|---|---|---|---|---|---|---|---|---|---|
| **Hit@1** | 0.0167 | 0.0421 | 0.0500 | 0.0000 | 0.1050 | 0.0025 | 0.0158 | 0.0125 | 0.0421 |
| **Hit@5** | 0.0514 | 0.1006 | 0.4188 | 0.0000 | 0.1975 | 0.1008 | 0.1708 | 0.0650 | 0.1736 |
| **Hit@10** | 0.0947 | 0.1033 | 0.4355 | 0.0000 | 0.3100 | 0.1144 | 0.2008 | 0.0800 | 0.1736 |
| **Hit@20** | 0.1047 | 0.1033 | 0.4355 | 0.0000 | 0.3150 | 0.1144 | 0.2008 | 0.0850 | 0.1736 |

| Agentless | astropy | dvc | ipython | pylint | scipy | sphinx | streamlink | xarray | geopandas |
|---|---|---|---|---|---|---|---|---|---|
| **Hit@1** | 0.0667 | 0.0400 | 0.0638 | 0.0426 | 0.1458 | 0.0833 | 0.0200 | 0.0612 | 0.0400 |
| **Hit@5** | 0.1111 | 0.1800 | 0.2979 | 0.1489 | 0.1875 | 0.2917 | 0.2400 | 0.2449 | 0.2400 |
| **Hit@10** | 0.1111 | 0.1800 | 0.3191 | 0.1489 | 0.1875 | 0.2917 | 0.2400 | 0.2449 | 0.2400 |
| **Hit@20** | 0.1111 | 0.1800 | 0.3191 | 0.1489 | 0.1875 | 0.2917 | 0.2400 | 0.2449 | 0.2400 |

Table 8: Joint-training Results and Cross-Repository Transfer Learning Analysis at the Func/Class granularity level.

| Joint-Type | | astropy | dvc | ipython | pylint | scipy | sphinx | streamlink | xarray | geopandas |
|---|---|---|---|---|---|---|---|---|---|---|
| Joint-ALL | **Hit@1** | 0.1796 | 0.2442 | 0.2178 | 0.2646 | 0.2647 | 0.1626 | 0.4276 | 0.1435 | 0.0999 |
| | **Hit@5** | 0.4379 | 0.5513 | 0.5041 | 0.5446 | 0.5603 | 0.5088 | 0.8170 | 0.4225 | 0.4140 |
| | **Hit@10** | 0.5953 | 0.6199 | 0.7175 | 0.6877 | 0.7272 | 0.6239 | 0.9012 | 0.5389 | 0.5635 |
| | **Hit@20** | 0.7268 | 0.7774 | 0.7858 | 0.7396 | 0.7949 | 0.6822 | 0.9408 | 0.6809 | 0.6955 |
| -w/o astropy | **Hit@1** | 0.1757 | 0.2268 | 0.2011 | 0.3056 | 0.2474 | 0.1741 | 0.3542 | 0.1410 | 0.0999 |
| | **Hit@5** | 0.4258 | 0.5773 | 0.5712 | 0.5565 | 0.5975 | 0.4608 | 0.8125 | 0.3986 | 0.3812 |
| | **Hit@10** | 0.5648 | 0.6478 | 0.7211 | 0.6633 | 0.7214 | 0.5979 | 0.8887 | 0.5195 | 0.5329 |
| | **Hit@20** | 0.6932 | 0.7916 | 0.7918 | 0.7375 | 0.8236 | 0.6713 | 0.9382 | 0.6562 | 0.7245 |
| -w/o astropy dvc | **Hit@1** | 0.1712 | 0.1466 | 0.1730 | 0.2677 | 0.2365 | 0.1841 | 0.4220 | 0.1390 | 0.0877 |
| | **Hit@5** | 0.4141 | 0.3940 | 0.4974 | 0.5798 | 0.5139 | 0.4909 | 0.8235 | 0.4014 | 0.3431 |
| | **Hit@10** | 0.5563 | 0.5505 | 0.7155 | 0.6717 | 0.6606 | 0.5965 | 0.8957 | 0.4932 | 0.4957 |
| | **Hit@20** | 0.6743 | 0.6776 | 0.7730 | 0.7688 | 0.7659 | 0.7394 | 0.9319 | 0.6138 | 0.7432 |
| -w/o astropy dvc ipython | **Hit@1** | 0.1748 | 0.1486 | 0.1853 | 0.2083 | 0.2364 | 0.1047 | 0.4359 | 0.1567 | 0.1026 |
| | **Hit@5** | 0.4146 | 0.4095 | 0.4385 | 0.5767 | 0.5201 | 0.4528 | 0.8246 | 0.3766 | 0.4003 |
| | **Hit@10** | 0.5413 | 0.5443 | 0.5628 | 0.6279 | 0.6412 | 0.5611 | 0.8859 | 0.4880 | 0.5864 |
| | **Hit@20** | 0.6638 | 0.6507 | 0.6973 | 0.6940 | 0.7507 | 0.6653 | 0.9193 | 0.6014 | 0.7080 |
| -w/o astropy dvc ipython pylint | **Hit@1** | 0.1742 | 0.1544 | 0.1888 | 0.2138 | 0.2312 | 0.2239 | 0.3435 | 0.1343 | 0.0961 |
| | **Hit@5** | 0.4162 | 0.4088 | 0.4624 | 0.4519 | 0.5460 | 0.4923 | 0.8017 | 0.3711 | 0.4165 |
| | **Hit@10** | 0.5504 | 0.5394 | 0.6020 | 0.5733 | 0.6917 | 0.5481 | 0.8937 | 0.4930 | 0.5457 |
| | **Hit@20** | 0.6769 | 0.6649 | 0.7131 | 0.6830 | 0.8043 | 0.6964 | 0.9505 | 0.6193 | 0.7600 |

Table 9: Ablation study on MPNN (File granularity level).

| Datasets | Metrics | GPS | GIN | GraphSage | GT | GAT |
|---|---|---|---|---|---|---|
| astropy | Hit@1 | 0.4155 | 0.4530 | 0.5024 | 0.1347 | 0.4873 |
| | Hit@5 | 0.8245 | 0.8810 | 0.8884 | 0.5575 | 0.8691 |
| | Hit@10 | 0.8909 | 0.9426 | 0.9428 | 0.7380 | 0.9458 |
| | Hit@20 | 0.9403 | 0.9766 | 0.9723 | 0.9081 | 0.9763 |
| dvc | Hit@1 | 0.3093 | 0.3270 | 0.3497 | 0.1604 | 0.4457 |
| | Hit@5 | 0.7601 | 0.6553 | 0.7184 | 0.5442 | 0.7412 |
| | Hit@10 | 0.8712 | 0.7828 | 0.8965 | 0.7159 | 0.8687 |
| | Hit@20 | 0.9394 | 1.0000 | 0.9798 | 0.8699 | 0.9798 |
| ipython | Hit@1 | 0.4349 | 0.5016 | 0.5631 | 0.2131 | 0.5439 |
| | Hit@5 | 0.8962 | 0.9564 | 0.9468 | 0.7369 | 0.9571 |
| | Hit@10 | 0.9744 | 0.9731 | 0.9795 | 0.9154 | 0.9769 |
| | Hit@20 | 1.0000 | 0.9962 | 1.0000 | 0.9538 | 0.9872 |
| pylint | Hit@1 | 0.2862 | 0.4327 | 0.3923 | 0.1869 | 0.5673 |
| | Hit@5 | 0.7003 | 0.8316 | 0.8367 | 0.6044 | 0.8333 |
| | Hit@10 | 0.8670 | 0.9209 | 0.9310 | 0.8098 | 0.9097 |
| | Hit@20 | 0.9444 | 0.9646 | 0.9848 | 0.9175 | 0.9747 |
| scipy | Hit@1 | 0.3893 | 0.5466 | 0.5321 | 0.2474 | 0.5728 |
| | Hit@5 | 0.8278 | 0.9305 | 0.8967 | 0.6407 | 0.9242 |
| | Hit@10 | 0.9180 | 0.9788 | 0.9725 | 0.8243 | 0.9732 |
| | Hit@20 | 0.9756 | 0.9867 | 0.9894 | 0.9493 | 0.9902 |
| sphinx | Hit@1 | 0.4889 | 0.5139 | 0.6369 | 0.1347 | 0.6389 |
| | Hit@5 | 0.8833 | 0.8833 | 0.9162 | 0.6319 | 0.9167 |
| | Hit@10 | 0.9531 | 0.9917 | 0.9917 | 0.8042 | 0.9750 |
| | Hit@20 | 0.9750 | 1.0000 | 1.0000 | 0.9167 | 1.0000 |
| streamlink | Hit@1 | 0.2993 | 0.7509 | 0.7133 | 0.3602 | 0.7079 |
| | Hit@5 | 0.7832 | 0.9857 | 0.9695 | 0.8477 | 0.9749 |
| | Hit@10 | 0.9695 | 0.9892 | 0.9892 | 0.9283 | 0.9910 |
| | Hit@20 | 0.9910 | 0.9892 | 0.9946 | 0.9857 | 0.9946 |
| xarray | Hit@1 | 0.3254 | 0.4399 | 0.4334 | 0.1451 | 0.4409 |
| | Hit@5 | 0.8087 | 0.8899 | 0.8438 | 0.5142 | 0.8315 |
| | Hit@10 | 0.9026 | 0.9723 | 0.9642 | 0.7235 | 0.9437 |
| | Hit@20 | 0.9637 | 0.9930 | 0.9930 | 0.9347 | 0.9912 |
| geopandas | Hit@1 | 0.3333 | 0.4462 | 0.5451 | 0.1562 | 0.4722 |
| | Hit@5 | 0.7812 | 0.8958 | 0.9062 | 0.5920 | 0.8681 |
| | Hit@10 | 0.9427 | 0.9670 | 0.9306 | 0.8090 | 0.9722 |
| | Hit@20 | 0.9896 | 0.9896 | 0.9688 | 0.9444 | 0.9792 |

Table 10: Ablation study on MPNN (Func/Class granularity level).

|  | Metrics | GPS | GIN | GraphSage | GT | GAT |
|---|---|---|---|---|---|---|
| astropy | Hit@1 | 0.1009 | 0.0382 | 0.0560 | 0.0276 | 0.0193 |
|  | Hit@5 | 0.2740 | 0.1327 | 0.1866 | 0.1067 | 0.1414 |
|  | Hit@10 | 0.4654 | 0.2238 | 0.2564 | 0.1835 | 0.3073 |
|  | Hit@20 | 0.5877 | 0.3493 | 0.4135 | 0.3486 | 0.4781 |
| dvc | Hit@1 | 0.1074 | 0.0042 | 0.0451 | 0.0760 | 0.0278 |
|  | Hit@5 | 0.4056 | 0.1292 | 0.1510 | 0.2042 | 0.1806 |
|  | Hit@10 | 0.5404 | 0.3024 | 0.3007 | 0.3063 | 0.2896 |
|  | Hit@20 | 0.6397 | 0.4777 | 0.5103 | 0.5044 | 0.4939 |
| ipython | Hit@1 | 0.0911 | 0.0321 | 0.0453 | 0.0000 | 0.0883 |
|  | Hit@5 | 0.3989 | 0.1670 | 0.1730 | 0.0775 | 0.2144 |
|  | Hit@10 | 0.6409 | 0.3371 | 0.2504 | 0.1879 | 0.3197 |
|  | Hit@20 | 0.7622 | 0.4342 | 0.5125 | 0.4490 | 0.5973 |
| pylint | Hit@1 | 0.1385 | 0.0448 | 0.0125 | 0.0094 | 0.0229 |
|  | Hit@5 | 0.3533 | 0.1531 | 0.1292 | 0.0490 | 0.2542 |
|  | Hit@10 | 0.4833 | 0.2281 | 0.2229 | 0.1094 | 0.4042 |
|  | Hit@20 | 0.6188 | 0.3760 | 0.3302 | 0.2950 | 0.5240 |
| scipy | Hit@1 | 0.1065 | 0.0750 | 0.0568 | 0.0231 | 0.0195 |
|  | Hit@5 | 0.2995 | 0.2364 | 0.2063 | 0.0963 | 0.1319 |
|  | Hit@10 | 0.4813 | 0.3227 | 0.2825 | 0.1887 | 0.2519 |
|  | Hit@20 | 0.6233 | 0.4618 | 0.5128 | 0.3170 | 0.4151 |
| sphinx | Hit@1 | 0.1232 | 0.0410 | 0.0565 | 0.0093 | 0.0993 |
|  | Hit@5 | 0.3213 | 0.1354 | 0.1479 | 0.0773 | 0.2068 |
|  | Hit@10 | 0.5178 | 0.2112 | 0.3202 | 0.0968 | 0.3547 |
|  | Hit@20 | 0.6569 | 0.3138 | 0.4576 | 0.2848 | 0.5058 |
| streamlink | Hit@1 | 0.1930 | 0.0830 | 0.0876 | 0.0771 | 0.1711 |
|  | Hit@5 | 0.5851 | 0.3412 | 0.2560 | 0.2638 | 0.3400 |
|  | Hit@10 | 0.7835 | 0.5564 | 0.4738 | 0.5158 | 0.5257 |
|  | Hit@20 | 0.8648 | 0.7671 | 0.7560 | 0.7468 | 0.7058 |
| xarray | Hit@1 | 0.0526 | 0.0133 | 0.0137 | 0.0155 | 0.0198 |
|  | Hit@5 | 0.2229 | 0.0929 | 0.0948 | 0.0665 | 0.1148 |
|  | Hit@10 | 0.3428 | 0.1789 | 0.1877 | 0.1488 | 0.2102 |
|  | Hit@20 | 0.4438 | 0.3038 | 0.3259 | 0.2460 | 0.3364 |
| geopandas | Hit@1 | 0.0645 | 0.0049 | 0.0412 | 0.0000 | 0.0020 |
|  | Hit@5 | 0.2136 | 0.0457 | 0.1089 | 0.0252 | 0.0909 |
|  | Hit@10 | 0.4097 | 0.2391 | 0.2349 | 0.0877 | 0.1829 |
|  | Hit@20 | 0.5585 | 0.3479 | 0.3593 | 0.2959 | 0.4079 |

