# OpenReview forum: "GREPO: A Benchmark for Graph Neural Networks on Repository-Level Bug Localization"
_ICLR.cc/2026/Conference — ICLR 2026 Conference Withdrawn Submission_

### Official Review · Reviewer_d85e · 2025-10-24

**Soundness:** 2
**Presentation:** 2
**Contribution:** 2
**Rating:** 2
**Confidence:** 3

**Summary:**

This paper introduces GREPO, a new benchmark dataset designed to facilitate research on repository-level bug localization using Graph Neural Networks (GNNs). The authors argue that existing methods, such as those based on Information Retrieval (IR) or Large Language Models (LLMs) with limited context, fail to adequately leverage the rich structural information inherent in code repositories. To address this, GREPO provides a large-scale dataset of 109 Python repositories, comprising over 10,000 bug-fixing pull requests, with the codebase pre-processed into graph structures suitable for direct use with GNNs.
The work's primary contribution is the benchmark itself, intended as a foundational resource for the community.

**Strengths:**

1.  The creation of a large, pre-processed, and graph-ready benchmark (GREPO) is a substantial engineering effort. Such a resource is valuable and can lower the barrier to entry for researchers wanting to apply GNNs to this domain.
2.  The pipeline for constructing the dataset, including the use of a temporal graph to efficiently handle different commit snapshots and the careful collection of high-quality labels from pull requests and issues, is well-designed.
3. The authors have made the code open-source.

**Weaknesses:**

1. The primary contribution of this work is to be the construction of a new dataset. Given that the main novelty lies in the dataset and its empirical findings, the paper might be a better fit for a conference with a dedicated "Datasets and Benchmarks" track or a top-tier software engineering venue (e.g., ICSE, FSE, ASE), where the contribution would be more prominently highlighted. For a venue like ICLR, the innovation seems limited.
2. The performance of the AgentLess baseline is perplexing and lacks sufficient analysis. It reportedly achieves a near-perfect 92.7% Hit@1 at the file level but a drastically poor 6.26% at the class/function level. This is highly counter-intuitive. The paper fails to discuss or acknowledge this strange result.

**Questions:**

1. The limitation discussed in Section 8 is critical. Could you provide more details about this experiment? What was the exact setup? Why do you believe the GNN's localization signal failed to help SWEBench?
2. The standard deviations reported in Table 1 are extremely high, in some cases larger than the mean itself (e.g., LocAgent). This severely weakens the claim of generalizability. Furthermore, could authors offer any insights into why the performance is so unstable?

---

### Official Review · Reviewer_2JMH · 2025-11-01

**Soundness:** 2
**Presentation:** 1
**Contribution:** 3
**Rating:** 2
**Confidence:** 4

**Summary:**

This paper presents GREPO, a large-scale benchmark dataset and evaluation suite for repository-level bug localization using Graph Neural Networks (GNNs). GREPO contains 109 Python repositories and over 10k bug-fixing pull requests, representing each repository as a temporal heterogeneous graph with directory, file, class, and function nodes and contain, call, and inherit edges. The authors build node and query features using LLM embeddings (Qwen3-Embedding-8B) and introduce anchor nodes and similarity features for subgraph extraction. Multiple GNN architectures (GIN, GraphSAGE, GAT, GPS, UniMP) are evaluated and compared against IR-based and LLM/agent baselines using Hit@k metrics. GREPO serves as a dataset and standardized evaluation pipeline intended to support future research into graph-based repository reasoning.

**Strengths:**

1.The research motivation is clear, and the overall idea is meaningful.
2.The ablation experiments validate the effectiveness of methodological design
3.GREPO provides a valuable dataset resource for graph-based software repository analysis

**Weaknesses:**

1.Unclear methodological exposition: The introduction clearly states the research motivation, but the subsequent sections fail to describe the motivation and construction process of the benchmark in a coherent way. As a result, the paper reads as fragmented and lacks a continuous narrative.
2.Definitions and formulas lack rigor: The concepts of anchor nodes, similarity features, and subgraph extraction are insufficiently described. In Figure 1 (page 4), anchor-related content is not mentioned at all in the “Dataset Construction” section (page 3), making the figure hard to interpret. Moreover, the definitions of anchor and similarity features are only provided later in Section 5 (“The GREPO Graph Formulation”), which disrupts the logical flow. In addition, the second formula on page 6 seems inconsistent with the surrounding text—please clarify this to ensure mathematical correctness.
3.Missing baselines: The authors criticize existing GNN methods (e.g., GNN-for-CFG, Huo et al., 2020; Ma & Li, 2022) in Section 2.2, but GREPO is not compared with these baselines in the experiments (Section 6.3).This omission weakens the validity of the empirical claims.
4.Lack of objectivity in the conclusions: In Section 6.3, the authors avoid discussing the fact that their GNN models are significantly outperformed by the Agentless (GPT-4o) baseline on the file-level Hit@1 metric (54.18 vs 92.72).In the Limitations section, they attribute poor results to issues with the agent framework instead of analyzing the root causes.This selective interpretation undermines the credibility of the conclusions.
5.Lack of interpretive depth: Experimental results are listed but not analyzed to yield deeper insights.The paper would benefit from qualitative case studies or error analyses to explain where and why the models succeed or fail.

**Questions:**

1.Could the authors provide a more detailed definition of anchor nodes and clarify their relationship to similarity features?
2.Were the GNN baselines mentioned in the introduction actually implemented or tested? If not, why were they omitted from the comparison?
3.Do the authors have any analysis or explanation for why GAT performs substantially worse than the Agentless baseline on some metrics?

---

### Official Review · Reviewer_Sb8k · 2025-11-03

**Soundness:** 3
**Presentation:** 3
**Contribution:** 2
**Rating:** 2
**Confidence:** 4

**Summary:**

The paper explores  Graph Neural Networks (GNNs) as an alternative to repository-level bug localization, motivated by their ability to model complex dependencies in codebases. For this, the authors introduce GREPO designed for repository-scale bug localization with GNNs, 109 Python repositories and over 10,000 bug-fixing pull requests. With experiments comparing various GNN architectures against existing baselines, the authors claim the strong potential of GNNs for the repository-level bug localization task.

**Strengths:**

1. The approach described in the paper to leverage GNNs for the bug localization task is very detailed and carefully designed.
2. The ablation experiments are pretty comprehensive and explore a variety of methodological choices.
3. The results, shown a subset of the GREPO benchmark, look very impressive.

**Weaknesses:**

While the experimental results presented by the authors do look very promising, below are reasons why I am not convinced yet of the author’s claims that GNNs will be able to outperform information retrieval (IR) approaches

1. **Inadequate Baselines**: While the authors claims to compare against IR approaches,  they haven’t considered any embedding-specific or retrieve-and-rerank style methods. I would suggest to compare against the SweRank approach [1], which uses a code embedding model for function retrieval and LLM for reranking.
2. **Limited Evaluation Benchmarks**: The paper only evaluates on the GREPO benchmark introduced in this paper. The authors should also evaluate the trained GNN models directly on the established Swe-Bench-Lite and LocBench benchmarks, for which there are extensive baseline results presented in [1].
3. **Novelty Claims**: The paper’s claim that they are the first to create a benchmark for graph-based bug localization is not actually true. LocAgent also released LocBench, which created a dependency-graph representation of the repo (although without temporal connections across commits) as done in this paper. The authors should acknowledge this accordingly.

If the above concerns are addressed in the rebuttal, I am happy to increase  my score.

[1] SweRank:  Software Issue Localization with Code Ranking; Reddy et al 2025.

**Questions:**

1. In Table 1, Agentless seems to have considerable high file-localization performance (>90 hit@1) but much lower function-level localization. Any explanation for this?
2. The paper released  109 repositories in GREPO but only use 9 of them evaluation. Is the scale of training the issue? And what measures were taken to validate the quality / correctness of the remaining 100 repositories
3. The function-level numbers for GAT seem to have very high (>10) confidence intervals. What could be the reason for such high variance?

---

### Note · Authors · 2026-02-14

I have read and agree with the venue's withdrawal policy on behalf of myself and my co-authors.

---

### Meta-Review · Area_Chair_EPHy · 2026-01-07

**Summary:**

Reviewers appreciate the effort in constructing a large, graph-ready benchmark for repository-level bug localization and acknowledge the relevance of the problem. However, they raise substantial concerns regarding the strength and completeness of the empirical evaluation, the adequacy of baselines, clarity of methodological exposition, and the validity of novelty claims. In particular, the evaluation is limited to a small subset of repositories, omits strong contemporary baselines and external benchmarks, and reports highly unstable results that are insufficiently analyzed. Several reviewers also note selective interpretation of results and weaknesses in presentation and narrative coherence. Collectively, these issues significantly weaken the paper’s claims and impact.

**Reviewer Concerns:**

No rebuttal was provided. All critical concerns remain unaddressed, including:

Missing strong baselines (e.g., embedding-based and retrieve-and-rerank methods) and a lack of evaluation on established benchmarks.

Limited evaluation scale (9/109 repositories) without sufficient justification or validation.

Overstated novelty claims relative to prior benchmarks (e.g., graph-based bug localization datasets).

High variance and counter-intuitive results without adequate explanation or analysis.

Unclear methodological descriptions, inconsistent presentation, and lack of qualitative or error analysis.

**Reviewer Scores:**

Given the absence of rebuttal and discussion, reviewers would be unlikely to increase their scores. I expect all reviewers to maintain their original ratings, which are consistently in the reject range.

---

### Decision · Program_Chairs · 2026-01-26

Reject